# Risk of Insulin Resistance and Metabolic Syndrome in Women with Hyperandrogenemia: A Comparison between PCOS Phenotypes and Beyond

**DOI:** 10.3390/jcm10040829

**Published:** 2021-02-18

**Authors:** Valentin Borzan, Elisabeth Lerchbaum, Cornelia Missbrenner, Annemieke C. Heijboer, Michaela Goschnik, Christian Trummer, Verena Theiler-Schwetz, Christoph Haudum, Roswitha Gumpold, Natascha Schweighofer, Barbara Obermayer-Pietsch

**Affiliations:** 1Department of Internal Medicine, Division of Endocrinology and Diabetology, Endocrinology Lab Platform, Medical University of Graz, Auenbruggerplatz 15, 8036 Graz, Austria; valentin.borzan@medunigraz.at (V.B.); elisabeth.lerchbaum@medunigraz.at (E.L.); cornelia.missbrenner@medunigraz.at (C.M.); michaela.goschnik@uniklinikum.kages.at (M.G.); christian.trummer@medunigraz.at (C.T.); verena.schwetz@medunigraz.at (V.T.-S.); christoph.haudum@medunigraz.at (C.H.); roswitha.gumpold@medunigraz.at (R.G.); natascha.schweighofer@medunigraz.at (N.S.); 2Center for Biomarker Research in Medicine (CBmed), 8010 Graz, Austria; 3Endocrine Laboratory, Department of Clinical Chemistry, University Medical Center, Vrije Universiteit, 1081 HV Amsterdam, The Netherlands; a.heijboer@amsterdamumc.nl

**Keywords:** PCOS, hyperandrogenism, free testosterone, insulin resistance, Rotterdam criteria

## Abstract

Polycystic ovary syndrome (PCOS) is the most common endocrine disorder in premenopausal women, with a wide spectrum of possible phenotypes, symptoms and sequelae according to the current clinical definition. However, there are women who do not fulfill at least two out of the three commonly used “Rotterdam criteria” and their risk of developing type 2 diabetes or obesity later in life is not defined. Therefore, we addressed this important gap by conducting a retrospective analysis based on 750 women with and without PCOS. We compared four different PCOS phenotypes according to the Rotterdam criteria with women who exhibit only one Rotterdam criterion and with healthy controls. Hormone and metabolic differences were assessed by analysis of variance (ANOVA) as well as logistic regression analysis. We found that hyperandrogenic women have per se a higher risk of developing insulin resistance compared to phenotypes without hyperandrogenism and healthy controls. In addition, hyperandrogenemia is associated with developing insulin resistance also in women with no other Rotterdam criterion. Our study encourages further diagnostic and therapeutic approaches for PCOS phenotypes in order to account for varying risks of developing metabolic diseases. Finally, women with hyperandrogenism as the only symptom should also be screened for insulin resistance to avoid later metabolic risks.

## 1. Introduction

Polycystic ovary syndrome (PCOS) is the most common endocrine disorder in women of childbearing age, affecting between 6 and 22 percent of all women world-wide depending on the definition [1,2]. Since the establishment of the Rotterdam consensus in 2003, PCOS has been defined by the presence of at least two out of three criteria: clinical or biochemical hyperandrogenism (HA), oligo- or amenorrhea (OM) and/or polycystic ovarian morphology (PCOM) [3]. In addition to these three central characteristics of PCOS, many women have several other comorbidities or consecutive diseases, including insulin resistance (IR) [4] with a high risk for the development of diabetes mellitus, low grade inflammation as well as dyslipidemia and obesity [5,6,7].

This definition results in several PCOS phenotypes, such as phenotype A (HA, OM, PCOM), phenotype B (HA, OM), phenotype C (HA, PCOM) and phenotype D (OM, PCOM). Many studies have shown that PCOS symptom severity as well as IR and other comorbidities occur mostly in women with HA phenotypes (A, B and C), while phenotype D shows a milder form of PCOS [1,8,9].

However, while women with two or more criteria are firmly diagnosed with PCOS, women with only one criterion are often disregarded or lost to follow-up, since they do not fit into the definition of PCOS. In cases where they exhibit HA without PCOM or OM, these women are often left without any diagnosis or treatment, despite the fact that HA has been shown to have adverse effects on several cardiovascular risk factors [10]. These women might not have actual menstrual irregularities or difficulties regarding their fertility, but they might still have metabolic risk factors associated with HA, which could lead to increased risk for type 2 diabetes, obesity, hypertension, dyslipidemia, metabolic syndrome or cardiovascular events like myocardial infarction or stroke later in life. 

The aim of this study is to examine the associations of HA without additional PCOS criteria in view of glucose and lipid metabolism. We hypothesized that women with HA only also have a higher risk of developing IR.

## 2. Methods

We conducted a cross-sectional study including 750 women between 18 and 45 years of age. They were recruited from 2007 to 2015 in the endocrine outpatient clinic of the Division of Endocrinology and Diabetology, University Hospital Graz. Most women were referred to our outpatient clinic by gynecologists or general practitioners for a detailed PCOS evaluation.

PCOS diagnosis was established with the presence of at least two Rotterdam crite-ria. Clinical HA was defined by the presence of hirsutism (modified Ferriman–Gallwey score (mFG score) > 4) [11], acne alopecia and/or seborrhea. Biochemical HA was defined by elevated total testosterone (TT > 0.77 ng/m), free testosterone (fTesto > 3.18 pg/mL), androstenedione (ASD > 3.2 ng/mL) and/or dehydroepiandrosterone-sulphate (DHEA-S > 2.75 ng/mL). OM was diagnosed with a mean cycle length of > 35 or < 25 days for the past 12 months or a single cycle length of > 90 days. PCOM was established via transvaginal ultrasound (TVU) by experienced gynecologists. In addition, other known causes of HA (such as Cushing’s syndrome, congenital adrenal hyperplasia, androgen secreting tumors or hyperprolactinemia) were excluded by testing for specific parameters (1 mg dexamethasone suppression test and salivary cortisol, where appropriate, 17α-hydroxyprogesterone (17OHP) and prolactin). Based on their PCOS criteria, all participants were assigned to six groups: four PCOS phenotypes A–D, a healthy control group and a sixth group including all women with only one Rotterdam criterion (“1RC”). 

Data and previous results from our PCOS cohort have already been published [12,13,14,15,16,17,18]. Nevertheless, data on women with only one criterion have not been published previously. 

The PCOS cohort study has been approved by the Institutional Review Board of the Medical University of Graz (EC 18-066 ex 06-07) and written informed consent was obtained from all patients prior to their recruitment.

The study methodology has already been published previously [15,18]. After admittance in our outpatient clinic, detailed patient histories were taken and physical examinations consisting of measurements of height, weight, systolic and diastolic blood pressure, pulse rate, waist and hip circumferences and clinical signs of HA were performed. Following an overnight fast, blood samples were taken for the measurement of hormone levels including thyroid-stimulating hormone (TSH), free thyroxine (fT4), free triiodothyronine (fT3), parathyroid hormone (PTH), 25-hydroxy-vitamin D (25OHD), basal cortisol, basal aldosterone and renin, luteinizing hormone (LH), follicle-stimulating hormone (FSH), 17α-estradiol, 17OHP, TT, fTesto, sex hormone binding globulin (SHBG), DHEA-S, ASD, ACTH, human growth hormone (HGH), prolactin as well as insulin-like growth factor 1 (IGF-1) and routine metabolic markers including a differential blood count, serum creatinine, electrolytes, gamma-glutamyl transferase (GGT), aspartate aminotransferase (AST), alanine aminotransferase (ALT), high-sensitivity C-reactive protein (hsCRP), glycated hemoglobin (HbA1c), total cholesterol, high-density lipoprotein (HDL), low-density lipoprotein (LDL) and triglycerides. In addition, most patients underwent a 2-h oral glucose tolerance test with 75 mg glucose dissolved in 300 mL of water and 4 timepoints of scheduled glucose and insulin measurements (baseline, 30 min, 60 min, 120 min). Homeostasis Model Assessment for Insulin Resistance (HOMA-IR) and Matsuda Indices were calculated to assess insulin resistance (IR) and sensitivity (IS), respectively [19]. IR was defined as HOMA-IR > 2 [20]. The presence of hyperglycemia and/or metabolic syndrome (MetS) was defined using the guidelines of the American Heart Association [21].

TSH, fT4 and fT3 as well as TT were measured using ADVIA Centaur^®^ Immunoas-says (Siemens Healthcare Diagnostics Inc., Tarrytown, NY, USA), PTH and SHBG using Elecsys 2010^®^ ECLIA (Roche Diagnostics GmbH, Mannheim, Germany), while 25OHD was assessed by an IDS iSYS 25OHD assay (Immunodiagnostic Systems Ltd., Boldon, UK) until January 2014 and by the IDS-iSYS-25OHDS assay from January 2014 onward. LH and FSH were measured using Access^®^ hLH and hFSH CLIA (Beckman Coulter Inc., Brea, CA, USA), respectively. 17β-estradiol and ASD were determined using IMMULITE^®^ CLIA assays (Siemens Healthcare Diagnostics Products Ltd., Glyn Rhonwy, UK), while 17OHP was measured by a 17OHP ELISA (IBL International GmbH, Hamburg, Germany). FTesto was measured using the ACTIVE^®^ Free Testosterone Radioimmunoassay (Immunotech s.r.o., Prague, Czech Republic), and DHEA-S by ELISA (Labor Diagnostika Nord, Nordhorn, Germany). HDL cholesterol, LDL cholesterol, total cholesterol, triglycerides and glucose levels were measured using Modular Analytics SWA (Roche Diagnostics GmbH, Mannheim, Germany). The summary of all assay reference ranges as well as intra- and inter-assay coefficients of variance can be found in Appendix A.

In a subsection of 113 patients, TT was measured using a well standardized liquid chromatography tandem mass spectrometry (LC-MS/MS) method at the Endocrine Laboratory, Department of Clinical Chemistry, VU University Medical Center, Amsterdam, the Netherlands, as described before [22,23]. Calculated free testosterone (CFT) was calculated based on the formula by Södergard et al. [24,25] and Passing & Bablok regression analysis was performed to compare the immunoassay results with LC-MS/MS results (immunoassay = −0.11 + 1.36 x LC-MS/MS). Both TT measurement methods correlated with a correlation coefficient of 0.81 (*p* < 0.001), while fTesto and CFT correlated with a coefficient of 0.73 (*p* < 0.001). The respective Passing & Bablok plots are added as Appendix A. Subsequently, all reported TT results in this paper were measured by immunoassay in order to include all 750 patients.

Data analysis was performed using SPSS version 25 (IBM, Armonk, NY, USA). Data in tables are presented as medians with interquartile ranges (IQR). Data distribution was evaluated using Kolmogorov–Smirnov and Shapiro–Wilk tests. One-way analysis of variance (ANOVA) and Welch-ANOVA were used for group comparisons for data with homogenous and non-homogenous variances, respectively, as well as Bonferroni and Games–Howell post-hoc tests, respectively. For comparisons within groups, Student’s *t*-tests and Mann–Whitney-U tests were performed in case of parametric and non-parametric data, respectively, and in case of two comparative subgroups. Whenever more subgroups were compared, ANOVA or Welch-ANOVA were used as described above. For all statistical tests, *p*-values < 0.05 were considered statistically significant.

In addition, odds ratios (OR) for the occurrence of IR, hyperglycemia and MetS were calculated (using dichotomous outcomes for all three diagnoses as dependent variables, respectively). ORs for IR were calculated using PCOS phenotypes as covariates, before and after adjusting for body mass index (BMI) and age. In a second step, instead of phenotypes, the presence of each elevated androgen (TT, fTesto, DHEA-S, ASD) as well as the other PCOS criteria (PCOM, OM) were used as covariates in order to assess the influence of each criterion. 

For hyperglycemia and MetS, we used BMI, the presence of elevated androgens (one dichotomous variable each for TT, fTesto, DHEA-S and ASD) as well as SHBG before and after adjusting for age and fasting insulin (I0) as covariates. I0 was chosen as an adjustment variable due to a potential influence of SHBG levels by insulin levels.

## 3. Results

### 3.1. Phenotype Characteristics

Out of 800 entries in our PCOS cohort, 750 were included in this study. The selection process is described in Figure 1. Of these patients, 652 women (87%) were diagnosed with PCOS, 75 women (10%) had only one Rotterdam criterion and 23 women (3%) had no criteria and were assigned to the control group. Of the PCOS women included, 392 (60%) had phenotype A, 170 (26%) had phenotype B, 52 (8%) had phenotype C and 38 (6%) had phenotype D. Descriptive statistics from anthropometric data and laboratory parameters are described in Table 1 and differences between the phenotypes, controls and women with one criterion (1RC) are shown in Figure 2.

Our results suggest a hierarchy between the phenotypes and groups in our study cohort according to signs of HA, with phenotypes A and B exhibiting the highest concentrations of androgens, followed by phenotype C and group 1RC, followed by phenotype D, with the control group always having the lowest androgen concentrations. Vice versa, SHBG levels showed an opposite trend, with the control group having the highest median value. 

A similar trend as for the androgen concentrations was found for BMI comparisons (Figure 2a). Phenotypes A and B had significantly higher BMI values than phenotype D (*p* = 0.006; *p* = 0.001, respectively), and phenotype B also showed a significant difference in BMI when compared to the control group (*p* = 0.016).

Thyroid and calcium metabolism parameters (TSH, fT_3_, fT_4_, 25OHD) were not found to be statistically different between all groups after post-hoc correction.

Glucose metabolism parameters such as HbA1c showed no statistical difference between the phenotypes and the groups as well. However, HOMA and Matsuda indices both displayed trends indicating that phenotypes A-C as well as group 1RC had higher values of HOMA and lower values of Matsuda indices, respectively, compared to the control group and phenotype D. In summary, these results were not statistically significant.

In addition, lipid metabolism parameters such as total cholesterol, HDL, LDL and triglycerides showed no statistical differences between the groups after post-hoc correction, but HDL showed a similar trend as SHBG or the Matsuda index.

### 3.2. Hyperandrogenemia Is Associated with Insulin Resistance in Women with One Rotterdam Criterion

The group with one Rotterdam criterion (group 1RC) can be further separated via the established Rotterdam characteristics: 4 women in the group had PCOM, 14 had OM and 57 had HA. Of the women with HA, 22 had clinical signs, 13 had hyperandrogenemia alone and 22 women had both clinical and biochemical HA (Figure 3).

Based on the differences between the phenotype groups, the 1RC and control groups, we tested our hypothesis that metabolic changes might also occur in women with only HA in the 1RC group. However, we could not find any significant differences in metabolic parameters based on HA status. In contrast, when the women were separated based on the hyperandrogenemia status, we found significant differences in LH, FSH, TT, fTesto, SHBG, DHEAS, ASD and the HOMA index (*p* = 0.042; *p* = 0.006; *p* = 0.005; *p* < 0.001; *p* = 0.010; *p* < 0.001; *p* < 0.001; *p* = 0.021, respectively), indicating that hyperandrogenemia is more indicative for metabolic changes in women than combined HA (Table 2).

### 3.3. Elevated Free Testosterone (fTesto) Levels Are Associated with an Increased Risk for Insulin Resistance Independently of PCOS Diagnosis and Phenotype

Based on the fact that IR was more prevalent in the 1RC women with hyperandrogenemia, we performed logistic regression analyses to assess the impact of the various phenotypes as well as any PCOS criteria on the risk of IR.

We found a significantly higher risk of developing IR compared to the control group in phenotypes A (OR 4.45, *p* = 0.017), B (OR 5.32, *p* = 0.009) and C (OR 4.89, *p* = 0.020). Phenotype D and the 1RC group also had ORs of 1.78 and 2.47, respectively, but they were not statistically significant. After adjusting for age and BMI, however, only age and BMI were significant (OR of 0.94, *p* = 0.001 and 1.20, *p* < 0.001, respectively), while none of the phenotypes showed significantly higher risks for developing IR compared to controls.

Next, we assessed the importance of the main serum androgens and the other Rotterdam criteria by defining the presence of elevated TT, fTesto, DHEA-S and ASD (one dichotomous variable for each androgen) as well as the presence of PCOM and OM as covariates. Before adjusting for age and BMI, elevated TT decreased the risk of developing IR significantly (OR 0.60, *p* = 0.020), whereas elevated fTesto levels increased the risk (OR 4.35, *p* < 0.001), while DHEA-S (OR 0.82), ASD (OR 1.31), PCOM (OR 0.97) and OM (OR 1.16) did not significantly impact IR risk. After adjusting for age and BMI, the ORs were 0.66 for TT (not significant n.s.), 1.95 for fTesto (*p* = 0.006), 0.71 for DHEA-S (n.s.), 1.81 for ASD (*p* = 0.006), 1.01 for PCOM (n.s.), 0.83 for OM (n.s.), 0.95 for age (*p* = 0.007) and 1.20 for BMI (*p* < 0.001). This indicates that fTesto might be the most indicative androgen associated with IR in women with hyperandrogenemia.

Since the 1RC group included both women with and without hyperandrogenemia (which might explain the not significant OR for phenotype analysis), we performed Spearman correlation analyses in women with HA comparing our four main glucose metabolism parameters (HOMA, Matsuda, glucose area-under-the-curve (AUC) and insulin AUC) with our hormones and metabolic parameters (Appendix A). The four glucose parameters significantly correlated with fTesto and inversely with SHBG. Next, we repeated the regression analysis above with the 1RC group and the control group only to point out the risk assessment of IR in women with one PCOS criterion compared to controls. Here, we found the following ORs for TT (0.25 (n.s.), fTesto (21.84, *p* = 0.013), DHEA-S (0.42, n.s.), ASD (2.15, n.s.), PCOM (5.07, n.s.) and OM (1.52, n.s.) before adjustment. After adjusting for age and BMI, only BMI was significant, with an OR of 1.17.

In contrast to IR, the risk of hyperglycemia per se was not dependent on androgen or SHBG levels, only BMI had a statistically significant influence with an OR of 1.10 (*p* < 0.001). This remained stable after adjustment for age and I0; BMI and age were statistically significant contributors for hyperglycemia, with ORs of 1.08 (*p* = 0.002) and 1.11 (*p* < 0.001), respectively.

This was not the case for MetS risk calculation, as the statistically significant unadjusted ORs were 2.04 (elevated fTesto, *p* = 0.034), 0.97 (SHBG, *p* = 0.007) and 1.19 (BMI, *p* < 0.001). After adjusting for age and I0, ORs were 0.96 (SHBG, *p* < 0.001), 1.12 (BMI, *p* < 0.001), 1.05 (age, *p* = 0.046) and 1.05 (I0, *p* < 0.001), while fTesto trended towards significance with an OR of 1.83 (*p* = 0.093).

## 4. Discussion

To our knowledge, this is the first publication to compare cardiovascular and metabolic risk factors in all four PCOS phenotypes as well as in women with only one Rotterdam criterion (group 1RC) and in control women without any Rotterdam characteristics.

Overall, our study results confirm the findings of a number of other groups in previous studies that the likelihood of developing metabolic risk is not similar for each phenotype [26,27]. In our cohort, phenotypes with HA (A-C) had higher average values of BMI and HOMA, while showing lower SHBG and Matsuda values. Previous studies by Polak et al. and Gupta et al. did not find any significant differences in IR prevalence [7,28]. However, this could be explained by a lower sample size (146 and 150, respectively, compared to 750 women included in our analysis) and their lack of Matsuda index calculation. Another study by Pergialiotis et al. found that varying cholesterol levels did not have an impact on the hormonal status and phenotypes of PCOS women [29]. These findings are confirmed by our results using ANOVA, showing no statistically significant differences between the PCOS phenotypes, the 1RC group and the control group based on total cholesterol, LDL or HDL.

In addition, we were able to use logistic regression modelling to show the importance of free testosterone as the most indicative androgen for an insulin resistance risk, even after the adjustment for age and BMI. A recent study by Antonio et al. similarly showed that free testosterone was associated with metabolic parameters, but not TT [30]. However, they did not demonstrate statistical significance after adjusting for BMI and age and the spectrum of PCOS phenotypes was not taken into account.

By showing a similar effect of androgen excess in the group with one Rotterdam criterion only (1RC), we were able to show that metabolic changes not only occur in women with two or more PCOS criteria, but also in women with hyperandrogenemia alone. Some previous studies have described a similar effect with regard to risk of diabetes or the metabolic syndrome [7,31]. However, they did not assess the early effect of developing IR, nor did they take into account the potential PCOS phenotypes using the Rotterdam criteria. Our study fills this void, as IR development is an important step in leading to impaired glucose tolerance or diabetes mellitus type 2 in later life [26].

There are several possible explanations for causes, whereby androgen excess can facilitate IR and diabetes. Many studies have shown that while many peripheral tissues develop IR, the theca cells in the ovaries remain sensitive to insulin. There, stimulation of the insulin receptors results in steroidogenesis and ultimately the secretion of testosterone. Thus, hyperinsulinemia leads to an oversecretion of androgens [32].

In addition, insulin stimulates the secretion of Gonadotropin-releasing hormone (GnRH) in the hypothalamus, resulting in pulsatile LH secretion typical of PCOS [33]. However, hyperinsulinism is not the primary cause of PCOS. Rosenfield et al. summarize in their review that the HA originating from ovarian dysfunction is the first step leading to the metabolic and hormonal changes seen in PCOS [32]. However, the pathomechanism of HA causing IR directly (even in non-obese women) is less understood.

In a recent study by Navarro et al., they could show in a mouse study that HA stimulates the β cells in the pancreas via the androgen receptors, leading to more excessive insulin secretion [34]. In addition, testosterone has been shown to affect the subcutaneous adipose tissue by downregulating the expression of hormone sensitive lipase (HSL), leading to decreased lipolysis [35]. Another study by Seow et al. demonstrated that HSL was also downregulated in visceral adipose tissue in PCOS women while the fatty acid transporter CD36 was upregulated [36]. Both circumstances might result in obesity and more IR in peripheral tissues. Despite the fact that none of these studies took into account the existence of different PCOS phenotypes, all have shown how androgen excess might impact glucose and lipid metabolism pathways, leading to IR. Therefore, a more differentiated view on the phenotype of PCOS is necessary to understand the underlying pathophysiology of the disorder.

In addition, knowledge on each phenotypes’ risk for metabolic sequelae might further help clinicians to decide on therapeutic options for women with PCOS-related signs, even with only few symptoms. Since the PCOS spectrum is very wide-spread, it is important to consider many different individual therapeutic approaches depending on a patient’s wish and circumstances as well as the clinical and biochemical markers in the respective case. Metformin as an additional medication beyond lifestyle modification has been used for many years as an off-label drug in women with PCOS, and it has established its place as a common treatment option in several guidelines [11,37,38,39].

After assessing our cohort data, we can conclude that women with hyperandrogenemia should be evaluated for lifestyle modification and/or metformin therapy, independent of the presence of any other PCOS criteria. Special consideration in this regard should be given to women demonstrating signs of IR or obesity during the check-up. Oral glucose tolerance tests might be performed when possible to assess IR, since baseline indices or insulin clamps are not always suitable for clinical routine [11].

Our cohort also showed several key differences to other PCOS cohorts and studies reported in the past. While our study showed a considerable percentage of women with IR, the prevalence of hyperglycemia or MetS was much lower as compared to other studies [10,31,40]. This could be explained by the age of our cohort participants, as 75 percent of the women were 30 years or younger. This aspect may therefore be of high interest for pubertal and very young women, presenting at their GPs or gynecological or endocrinological specialists.

Notwithstanding our findings, this study has several limitations that need to be addressed: Despite the overall sample size of 750 women, the sample sizes for the various phenotypes differ due to the recruitment nature of the PCOS cohort as well as the phenotype distribution. Phenotypes C, D and especially the control group have a comparatively smaller sample size. As a result, some of our parameters showing no significant differences between the groups might have been significantly different with larger sample sizes in these groups, especially after post-hoc correction. Even statistically significant results need to be interpreted carefully as a result of this discrepancy. Recruitment for more cohort patients with all phenotypes including C, D, 1RC and control women is ongoing in order to increase our sample size and to obtain additional results.

Another consequence of the sample size is the age difference between the control group and the other five groups. Since we recruit our participants from our outpatient clinic, a large part of patients presents with at least one Rotterdam criterion (as the reason for their visit). This presents a significant preselection bias, making any inference of the prevalence of PCOS phenotypes and symptoms unreliable. As a result, recruiting women between the ages of 20 and 30 years without any PCOS criteria proved quite difficult.

An additional limitation of this cohort is the lack of overall LC-MS/MS-based hormone measurements. We performed a Passing & Bablok regression to compare our immunoassay results with LC-MS/MS results in a defined subgroup of 113 participants to show the possible effect of the used methodology. In addition, free testosterone was not measured using equilibrium dialysis; the results were based on an established radioimmunoassay.

Summarizing our findings, we were able to show elevated free testosterone as an independent risk factor associated with the development of insulin resistance in both women with PCOS and in women with only one Rotterdam criterion (1RC). Women with hyperandrogenemia in the 1RC group should not be dismissed from clinics and private practices but instead be considered for lifestyle modification and/or metformin therapy in case of elevated insulin levels or impaired glucose tolerance, because of their high risk for insulin-associated disturbances.

## 5. Conclusions

We can draw several conclusions from this study:It is important to distinguish between the various PCOS phenotypes, as they could impact therapy decisions and potential later risk for metabolic diseases.Free testosterone was the most indicative androgen for the development and prevalence of insulin resistance and potential later progression to a metabolic syndrome/impaired glucose tolerance.Women who do not meet at least two Rotterdam criteria should still be screened for androgen excess, as they also have an increased risk for developing insulin resistance. In case IR is present, lifestyle modification and/or metformin therapy should be considered.

## Figures and Tables

**Figure 1 jcm-10-00829-f001:**
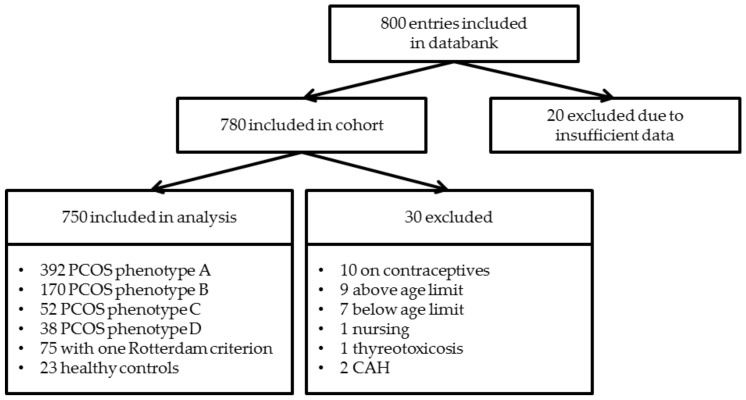
Flowchart highlighting the selection process and criteria used for including participants in the study. PCOS: Polycystic ovary syndrome; CAH: Congenital Adrenal Hyperplasia.

**Figure 2 jcm-10-00829-f002:**
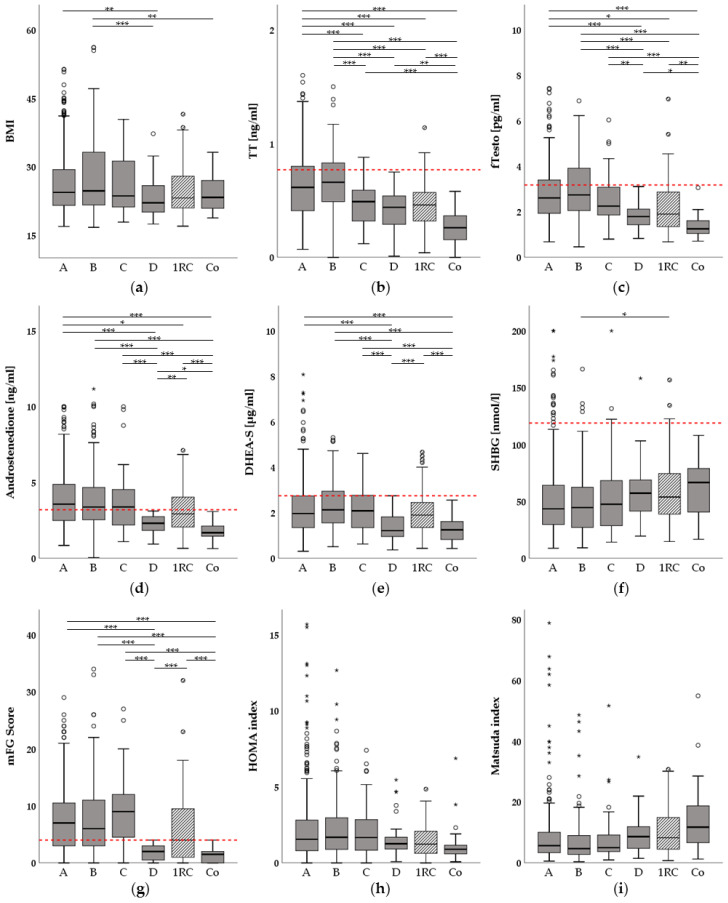
Boxplots showing hormonal and metabolic differences between the polycystic ovary syndrome (PCOS) phenotypes A–D and the one Rotterdam criterion (1RC) and control groups. *: *p* < 0.05; **: *p* < 0.01; ***: *p* < 0.001; Red horizontal lines depict upper reference limits for the respective parameter according to the reference laboratory; Circles in the figure depict outliers, stars above the colums represent extreme values; (**a**) Differences in body-mass-index (BMI); (**b**) Differences in total testosterone (TT, measured via immunoassay); (**c**) Differences in free testosterone (fTesto); (**d**) Differences in androstenedione (ASD); further ASD outliers were: 22.05 ng/mL (group A) and 24.70 ng/mL (group B); (**e**) Differences in dehydroepiandrosterone-sulphate (DHEA-S) (**f**) Differences in sex-hormone binding globulin (SHBG); (**g**) Differences in modified Ferriman-Gallwey (mFG) score; (**h**) Differences in homeostasis model-assessment for insulin resistance (HOMA-IR); further HOMA-IR outliers were: 29.21 (group A); 25.96 (group A); 22.67 (group B); 17.76 (group B) and 17.30 (group 1RC); (**i**):Differences in Matsuda index; further Matsuda outliers were: 135.26 (group A) and 145.07 (group B); Phenotype definitions: A: Hyperandrogenism (HA), Oligomenorrhea (OM) and Polycystic Ovarian Morphology (PCOM) present; B: HA and OM present; C: HA and PCOM present; D: OM and PCOM present; 1RC: Only one criterion (HA, OM or PCOM) present; Co: Control group (no Rotterdam criteria present).

**Figure 3 jcm-10-00829-f003:**
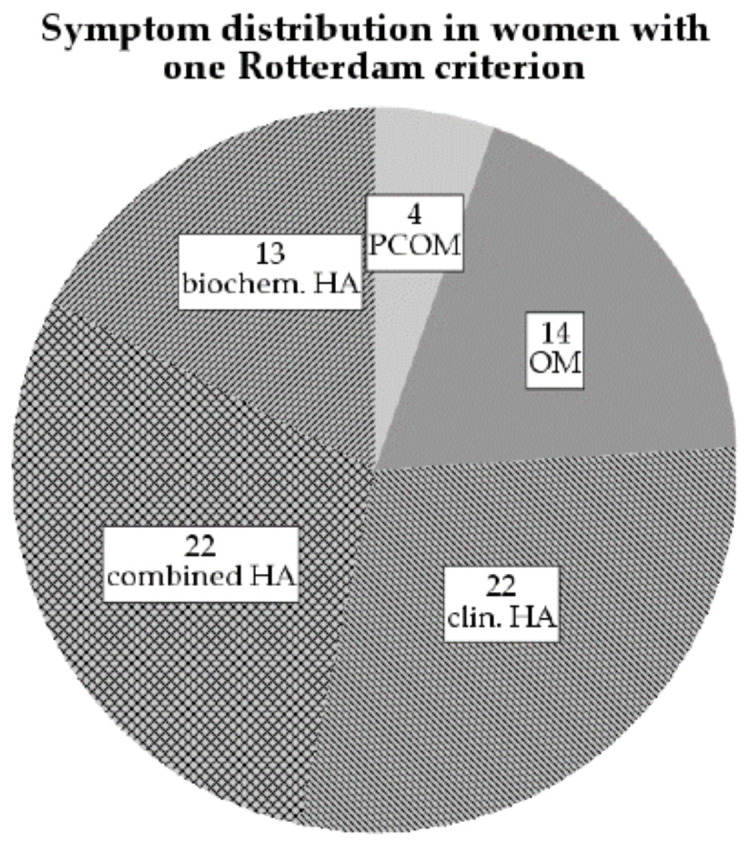
Pie chart showing the symptom distribution among 75 women with only one Rotterdam criterion (group 1RC); PCOM: Polycystic ovarian morphology; OM: Oligo-/Amenorrhea; HA: Hyperandrogenism; biochem.: biochemical; clin.: clinical.

**Table 1 jcm-10-00829-t001:** Characteristics of polycystic ovary syndrome (PCOS) phenotype A–D, patients with one Rotterdam criterion (1RC) and healthy controls.

Parameter	A	B	C	D	1RC	Co	*p*
*n*	392	170	52	38	75	23	-
Age (years)	26.6	27.6	27.3	27.6	27.6	36.7	<0.001
	(22.8–30.2)	(23.2–31.1)	(23.9–29.5)	(23.1–30.4)	(22.5–33.9)	(31.6–40.9)	
BMI (kg/m^2^)	24.4	24.7	23.6	22.1	23.1	23.3	<0.001
	(21.5–29.4)	(21.6–33.2)	(21.0–31.4)	(19.9–26.3)	(20.9–28.2)	(20.8–27.2)	
TT (ng/mL)	0.62	0.66	0.49	0.44	0.46	0.26	<0.001
	(0.41–0.80)	(0.49–0.83)	(0.32–0.59)	(0.29–0.55)	(0.32–0.58)	(0.12–0.37)	
fTesto (pg/mL)	2.61	2.74	2.25	1.79	1.90	1.25	<0.001
	(1.93–3.40)	(2.05–3.93)	(1.86–3.14)	(1.43–2.16)	(1.34–2.90)	(1.02–1.67)	
ASD (ng/mL)	3.57	3.38	3.39	2.31	2.93	1.69	<0.001
	(2.49–4.89)	(2.52–4.68)	(2.20–4.54)	(1.79–2.81)	(2.05–4.08)	(1.37–2.14)	
DHEA-S (µg/mL)	1.97	2.13	2.09	1.22	1.90	1.25	<0.001
	(1.34–2.75)	(1.55–2.96)	(1.35–2.78)	(0.94–1.83)	(1.34–2.48)	(0.78–1.67)	
SHBG (nmol/L)	43.6	44.6	47.6	57.3	53.9	66.7	0.023
	(29.8–64.3)	(27.1–62.5)	(28.4–68.7)	(41.4–69.7)	(38.3–77.4)	(39.5–80.4)	
mFG score (1)	7	6	9	2	4	1.5	<0.001
	(3–11)	(3–11)	(4–12)	(0–3)	(1–10)	(0–2)	
LH (mIU/mL)	9.28	8.24	6.40	9.22	5.87	3.53	0.038
	(5.29–14.00)	(4.49–13.10)	(4.64–13.21)	(3.76–12.75)	(3.24–9.31)	(2.41–8.18)	
FSH (mIU/mL)	5.60	5.57	5.41	6.48	4.91	7.84	0.143
	(4.27–7.06)	(4.05–7.04)	(3.80–7.60)	(5.45–8.03)	(3.35–7.15)	(4.34–9.94)	
LH/FSH ratio (1)	1.64	1.43	1.51	1.40	1.12	0.71	<0.001
	(1.07–2.46)	(0.99–2.23)	(0.84–1.96)	(0.88–1.75)	(0.76–1.85)	(0.40–1.18)	
TSH (µIU/mL)	1.92	1.80	1.92	1.69	1.77	1.65	0.38
	(1.41–2.61)	(1.23–2.39)	(1.46–2.87)	(1.02–2.48)	(1.23–2.42)	(1.16–2.09)	
fT_4_ [pmol/L]	14.2	14.5	14.2	14.1	14.6	14.7	0.868
	(12.8–15.8)	(13.3–15.7)	(12.9–15.2)	(12.4–16.1)	(13.0–16.2)	(13.3–16.0)	
fT_3_ [pmol/L]	5.0	5.0	5.0	4.9	4.8	4.5	<0.001
	(4.6–5.4)	(4.6–5.3)	(4.7–5.4)	(4.4–5.1)	(4.4–5.2)	(4.4–5.0)	
Prolactin (ng/mL)	9.4	10.0	10.4	8.9	10.4	9.2	0.847
	(7.5–12.9)	(7.8–14.1)	(8.2–13.3)	(5.6–11.9)	(7.9–15.7)	(7.6–15.7)	
25OHD (ng/mL)	26.1	25.3	23.1	25.9	24.4	26.3	0.746
	(18.5–33.1)	(16.8–31.8)	(19.0–32.0)	(17.9–30.7)	(17.9–33.3)	(18.9–34.9)	
HbA1c (mmol/moL)	33	33	33	31	33	34	0.324
	(31–35)	(31–35)	(31–36)	(30–33)	(31–34)	(31–37)	
HOMA-IR (1)	1.6	1.7	1.7	1.3	1.2	0.9	0.215
	(0.8–2.8)	(0.9–3.0)	(0.8–2.8)	(0.9–1.7)	(0.6–2.1)	(0.5–1.3)	
Matsuda (1)	5.7	4.7	5.0	8.6	8.3	11.8	0.198
	(3.3–10.1)	(2.8–9.1)	(3.7–9.9)	(4.8–12.1)	(4.5–15.1)	(6.3–21.1)	
IR present (*n* (%))	157 (40.1)	75 (44.4)	22 (42.3)	8 (21.1)	20 (27.0)	3 (13.0)	-
Hyperglycemia present (*n* (%))	25 (6.4)	11 (6.5)	2 (3.8)	2 (5.3)	5 (6.7)	3 (13.0)	-
Total cholesterol (mg/dL)	175	175	167	173	174	179	0.403
	(155–197)	(156–199)	(153–189)	(155–198)	(155–196)	(164–198)	
HDL (mg/dL)	63	62	61	74	69	67	0.082
	(52–75)	(52–77)	(51–74)	(61–82)	(57–83)	(57–82)	
LDL (mg/dL)	95	95	89	86	92	101	0.119
	(78–116)	(81–120)	(72–107)	(71–113)	(74–111)	(94–121)	
Triglycerides (mg/dL)	72	78	70	64	59	58	0.020
	(54–98)	(56–101)	(52–88)	(45–78)	(48–75)	(45–71)	
MetS present (*n* (%))	57 (14.5)	25 (14.7)	5 (9.6)	2 (5.3)	5 (6.7)	1 (4.3)	-

The *p*-value in the last column shows the overall ANOVA/Welch-ANOVA statistical significance; BMI: Body mass index; TT: Total testosterone (measured via immunoassay); fTesto: Free testosterone; ASD: Androstenedione; DHEA-S: Dehydroepiandrosterone-sulphate; SHBG: Sex-hormone binding globulin; mFG score: Modified Ferriman–Gallwey score; LH: Luteinizing hormone; FSH: Follicle-stimulating hormone; TSH: Thyreoid-stimulating hormone; fT4: free thyroxine; fT3; free triiodothyronine; 25OHD: 25-hydroxy-vitamin D: HbA1c: Glycated hemoglobin; HOMA-IR: Homeostasis model assessment for insulin resistance; IR: Insulin resistance; HDL: High-density lipoprotein; LDL: Low-density lipoprotein; MetS: Metabolic Syndrome; Phenotype definitions: A: Hyperandrogenism (HA), Oligomenorrhea (OM) and Polycystic Ovarian Morphology (PCOM) present; B: HA and OM present; C: HA and PCOM present; D: OM and PCOM present; 1RC: Only one criterion (HA, OM or PCOM) present; Co: Control group (no Rotterdam criteria present).

**Table 2 jcm-10-00829-t002:** Group 1RC (women with only one Rotterdam criterion) group characteristics and comparisons.

Group Comparison by:	Hyperandrogenemia		
Parameter	Yes	No	*p*
*n*	35	40	-
Age (years)	27.6 (22.5–33.9)	36.7 (31.6–40.9)	0.274
BMI (kg/m^2^)	23.1 (20.9–28.2)	23.3 (20.8–27.2)	0.530
TT (ng/mL)	0.46 (0.32–0.58)	0.26 (0.12–0.37)	0.005
fTesto (pg/mL)	1.90 (1.34–2.90)	1.25 (1.02–1.67)	<0.001
ASD (ng/mL)	2.93 (2.05–4.08)	1.69 (1.37–2.14)	<0.001
DHEA-S (µg/mL)	1.90 (1.34–2.48)	1.25 (0.78–1.67)	<0.001
SHBG (nmol/L)	53.9 (38.3–77.4)	66.7 (39.5–80.4)	0.010
mFG Score (1)	4 (1–10)	1.5 (0–2)	0.656
LH (mIU/mL)	5.87 (3.24–9.31)	3.53 (2.41–8.18)	0.042
FSH (mIU/mL)	4.91 (3.35–7.15)	7.84 (4.34–9.94)	0.006
HOMA-IR (1)	1.2 (0.6–2.1)	0.9 (0.5–1.3)	0.021
Matsuda (1)	8.3 (4.5–15.1)	11.8 (6.3–21.1)	0.092
Total cholesterol (mg/dL)	174 (155–196)	179 (164–198)	0.231*
HDL (mg/dL)	69 (57–83)	67 (57–82)	0.910*
LDL (mg/dL)	92 (74–111)	101 (94–121)	0.072*
Triglycerides (mg/dL)	59 (48–75)	58 (45–71)	0.884*

*p*-values were derived from Mann–Whitney U tests; *: Student’s *t*-test instead of Mann–Whitney U test was used for analysis due to normally distributed data; BMI: Body mass index; TT: Total testosterone (measured via immunoassay); fTesto: Free testosterone; ASD: Androstenedione; DHEA-S: Dehydroepiandrosterone-sulphate; SHBG: Sex-hormone binding globulin; mFG score: Modified Ferriman–Gallwey score; LH: Luteinizing hormone; FSH: Follicle-stimulating hormone; HOMA-IR: Homeostasis model assessment for insulin resistance; HDL: High-density lipoprotein; LDL: Low-density lipoprotein.

## Data Availability

The data presented in this study are available on request from the corresponding author. The data are not publicly available due to ethical and privacy reasons.

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
