# Peer review of "Risk of Insulin Resistance and Metabolic Syndrome in Women with Hyperandrogenemia: A Comparison between PCOS Phenotypes and Beyond"

_jcm, 2021, doi:10.3390/jcm10040829_

Round 1

Reviewer 1 Report

The authors investigated in the present study the "Risk of Insulin Resistance and Metabolic Syndrome in Women with Hyperandrogenemia". The mansucript is of scientific importance; however, certain points further clarification prior to its publication.

1) The authors mention inside the abstract that "hyperandrogenemia is a driving factor for developing insulin resistance also in women with no other Rotterdam criterion" and also mention inside their manuscript that "Hyperandrogenemia is the driving factor for insulin resistance"; however, it is already known that insulin resistance is actually the driving factor for hyperandrogenemia and that correctio of hyperinsulinism with metformin may help restore androgen to normal on nearly normal levels. In their manuscript they compared the whole 1RC group with controls but not women with hyperandrogenemia only with the control group (57 vs 23); hence, the abstract`s statement is poorly understood. 

2) The impact of mild hypercholesterolemia on glycemic and hormonal profiles of PCOS women has been already mentioned (2018 Mar 29;34(3):/j/hmbci.2018.34.issue-3/hmbci-2018-0002/hmbci-2018-0002.xml.) and the authors should elaborate on the findings of this previous study and compare them to their findings.

3) In the specific subsection of women with HA only i would like the authors to perform correlation analysis with insulin resistance markers

4)Cut-off values for metabolic and hormonal normal values should be provided even in the form of supplemental material for reproducibility purposes, coefficients of variation for ELISA parameters should be also provided.

5) It is unclear to me if the authors used equilibrium dialysis for the free-testo assessment and given the importance of this index in their study i believe that they should specify this, as the assessment of this hormone with the analog method is misleading.

Author Response

Reply on manuscript 1070775 to Reviewer 1:
The authors investigated in the present study the "Risk of Insulin Resistance and Metabolic Syndrome
in Women with Hyperandrogenemia". The mansucript is of scientific importance; however, certain points further clarification prior to its publication.

Point 1) The authors mention inside the abstract that "hyperandrogenemia is a driving factor for
developing insulin resistance also in women with no other Rotterdam criterion" and also mention inside their manuscript that "Hyperandrogenemia is the driving factor for insulin resistance"; however, it is already known that insulin resistance is actually the driving factor for hyperandrogenemia and that correctio of hyperinsulinism with metformin may help restore androgen to normal on nearly normal levels. In their manuscript they compared the whole 1RC group with controls but not women with hyperandrogenemia only with the control group (57 vs 23); hence, the abstract`s statement is poorly understood.

Response: Thank you for your comment on this topic. We acknowledge that PCOS pathophysiology is
more complex than suggested by the sentences you mentioned. As a result, we have changed the, respective sentences in the abstract (page 1, line 25) and heading of chapter 3.2 (page 8, line 213). In addition, we have added three paragraphs in the discussion to elaborate on the pathomechanisms and
interactions between hyperandrogenism and insulin resistance (pages 11-12, paragraphs 5-7, lines 315-338).

Point 2) The impact of mild hypercholesterolemia on glycemic and hormonal profiles of PCOS women has been already mentioned (2018 Mar 29;34(3):/j/hmbci.2018.34.issue-3/hmbci-2018-0002/hmbci2018-0002.xml.) and the authors should elaborate on the findings of this previous study and compare
them to their findings.

Response: Thank you for suggesting this study. We have incorporated this important publication in our
discussion and compared the results with ours regarding lipid metabolism (page 11, paragraph 2, lines 295-300).
Point 3) In the specific subsection of women with HA only i would like the authors to perform correlation analysis with insulin resistance markers
2

Response: We thank the reviewer for suggesting this method. As we were not sure about the
parameters you wanted to be specifically tested, we performed a correlation analysis between our
main IR markers (HOMA, Matsuda, GlucoseAUC and InsulinAUC) with our main hormonal and metabolic markers in the subsection of 1RC women with clinical and biochemical HA. The results are
provided as supplementary table 2. We have incorporated the main findings of the analysis on page 10, paragraph 4, lines 265-270. We hope this is satisfactory to you.

Point 4) Cut-off values for metabolic and hormonal normal values should be provided even in the form of supplemental material for reproducibility purposes, coefficients of variation for ELISA parameters should be also provided.

Response: We agree with reviewer 1 and have uploaded a supplementary table (supplementary table1) listing the analytical assays used in this study, specifying the parameters that were asked for. Thank you for your suggestion.

Point 5) It is unclear to me if the authors used equilibrium dialysis for the free-testo assessment and given the importance of this index in their study i believe that they should specify this, as the assessment of this hormone with the analog method is misleading.

Response: Thank you for this comment. As described on page 3 of our manuscript, paragraph 4, line 116, our free testosterone measurements were performed using the ACTIVE® Free Testosterone Radioimmunoassay (Immunotech s.r.o., Prague, Czech Republic). In addition, we have added this circumstance in our discussion section. We hope that this is a satisfactory answer to your comment.
We hope that our answers have addressed all comments satisfactorily and nothing more stands in the way of an acceptance of this manuscript.

Yours sincerely,
B. Obermayer-Pietsch and V. Borzan

Reviewer 2 Report

This is a very interesting and – in my eyes – clinically relevant study about an underrated topic, namely the relevance of hyperandrogenemia with and without PCOS. The study focuses on parameters of metabolism in women with only one PCOS criterion for the first time. I have two major concerns which need to addressed. Thus, I recommend major revisions.

Major comments:

  • Why does the manuscript head say “Int. J. Environ. Res. Public Health 2020”? Has this manuscript been submitted to the Int. J. Environ. Res. Public Health previously or is it under review there these days?
  • Methods: “The PCOS cohort study has been approved by the Institutional Review Board of the Medical University of Graz (EC 18-066 ex 06-07) and written informed consent was ob-tained from all patients prior to their recruitment.” – what about the “1RC” group? Was there an IRB approval for this study? Since the “1RC” group was not part of the prospective studies, was this a retrospective study? IF this is the case, the retrospective data acquisition in one study group needs to be addressed as a study limitation.

Minor comments:

  • “OM” should be defined at its first appearance in the main text.
  • Introduction: “The aim of this study is to examine the associations of hyperandrogenemia with-out additional PCOS […]” – should be “The aim of this study is to examine the associations of HA without additional PCOS […]”
  • A few typos/formatting errors should be corrected throughout the manuscript: “de-fined”, “estab-lished” and so on.

Author Response

Reply on manuscript 1070775 to Reviewer 2:

This is a very interesting and – in my eyes – clinically relevant study about an underrated topic, namely the relevance of hyperandrogenemia with and without PCOS. The study focuses on parameters of metabolism in women with only one PCOS criterion for the first time. I have two major concerns which need to addressed. Thus, I recommend major revisions.

Response: We thank you for your kind words and the opportunity to address the questions raised.

Why does the manuscript head say “Int. J. Environ. Res. Public Health 2020”? Has this manuscript been submitted to the Int. J. Environ. Res. Public Health previously or is it under review there these days?

Response: When we submitted our manuscript, we used the template found on the JCM-homepage. Our submitted version did not mention this journal, nor should it mention it. This submission is our first one of these data. It has not been submitted nor considered for publication at the “Int. J. Environ. Res. Public Health 2020” in the past or present, and we cannot explain why that journal would be mentioned in the edited version of our manuscript. With this cover letter, we are also submitting our original manuscript (version 1.0 submitted on December 25th 2020) as well as our updated (edited) version (version 1.1 from February 7th 2021) to prove this point. In both versions, you will find the correct journal name at present. We thank you for pointing this out.

Methods: “The PCOS cohort study has been approved by the Institutional Review Board of the Medical University of Graz (EC 18-066 ex 06-07) and written informed consent was ob-tained from all patients prior to their recruitment.” – what about the “1RC” group? Was there an IRB approval for this study? Since the “1RC” group was not part of the prospective studies, was this a retrospective study? IF this is the case, the retrospective data acquisition in one study group needs to be addressed as a study limitation.

Response: Thank you for asking this question. All women from the 1RC and control groups were also recruited as participants in our PCOS cohort. In most cases, women were recruited during their stay at our outpatient clinic, and quite often we had no prior records regarding PCOS criteria. That meant that not all criteria could be evaluated while the patient was still at the clinic, since laboratory analyses took some time. Thus, patients were often recruited with uncertain PCOS status, and only after the lab analyses were completed, we completed data entry into our cohort files. In other cases, some PCOS criteria had to be reevaluated or redefined after their outpatient’s clinic visit (for example in case, the 2

patients were referred to a gynecologist for an ultrasound checkup afterwards and were found to have/not have PCOM etc.). Data of women who turned out not to have the relevant PCOS criteria have not be used for any prospective PCOS studies resulting from our cohort recruitment, however they were still part of our cohort and their inclusion was covered by this IRB approval. We used the original clinical and laboratory data from all our patients. For this manuscript, no retrospective analyses (i.e. no new tests or repeated laboratory measurements) were performed. Therefore, we did not seek any new IRB approval. We hope that this answers your question regarding IRB approval of the 1RC group satisfactorily.

“OM” should be defined at its first appearance in the main text.

We would like to draw your attention towards the first paragraph of our introduction chapter on page 2 of the manuscript, line 5: “clinical or biochemical hyperandrogenism (HA), oligo- or amenorrhea (OM) […]”. This was our first mention of this abbreviation.

Introduction: “The aim of this study is to examine the associations of hyperandrogenemia with-out additional PCOS […]” – should be “The aim of this study is to examine the associations of HA without additional PCOS […]”

Thank you, we have adapted this section accordingly. (page 2, line 56)

A few typos/formatting errors should be corrected throughout the manuscript: “de-fined”, “estab-lished” and so on.

Thank you for pointing this out. We have noticed several hyphenated words throughout our manuscript. However, we believe this to be an editing issue, since we did not hyphenate these words in our originally submitted version. We hope to be able to contact the editing team to address this issue with them directly. We have corrected all hyphenated words that we could find.

We hope that our answers have addressed all comments satisfactorily and nothing more stands in the way of an acceptance of this manuscript.

Yours sincerely,

  1. Obermayer-Pietsch and V. Borzan

Reviewer 3 Report

The manuscript entitled „Risk of Insulin Resistance and Metabolic Syndrome in Women with Hyperandrogenemia – a comparison between PCOS phe-notypes and beyond” by Borzan et al. is generally well written and structured in a reasonable way. 

The main conception of the work seems reasonable. The effects of hyperandrogenism on metabolic abnormalities in patients with PCOS are widely known (as summarized e.g. by Sanchez-Garrido and Tena-Sempere (2020). Metabolic dysfunction in polycystic ovary syndrome: Pathogenic role of androgen excess and potential therapeutic strategies. Molecular metabolism, 35, 100937.). Also the question of the metabolic effects of hyperandrogenism in postmenopausal women is well-desribed in literature. In turn, the risk of development of the insulin resistance in hyperandrogenic premenopausal women needs a clear evaluation, which is the main aim of the submitted work.

Of course, some of the obtained results could be anticipated a priori. Nevertheless, it is important to evidently show the clinical results supporting the predicted relations, which is provided in this work.

Some specific remarks:

  1. Sometimes formatting looks strange especially by hyphenation e.g. charac-teris-tics, dis-eases.
  2. There is a huge disparity between the sizes of control or test groups and the PCOS ones (for the data presented in Tab.1). One should make the inferences cautiously and ponder whether they are really statistically significant; especially when the data describing the phenotype A of PCOS (n = 392) are compared with almost all other groups (n<78)  - in that case the sizes of control/other groups (except from PCOS B) are below 20% of the PCOS A group. I understand that the statistical tests confirmed the validity of the obtained relations between the investigated parameters. Still, the disproportions in sample sizes are can be a problematic issue.
  3. I think that the possible explanation of the observed relation between the free testosterone level and risk of developing metabolic type 2 diabetes and obesity should be provided taking into account the mechanistic link between them. In Discussion the Authors may provide the anticipated mechanism of how the androgens can imply the liability of patients to metabolic disorders. For example, the Authors can mention that androgens can downregulate hormone-sensitive lipase (HSL) and β2 adrenergic receptor expression etc.
  4. Could you comment on the relative amount of outliers in each group. Is it highest in PCOS A group? How can it be explained?

Author Response

Reply on manuscript 1070775 to Reviewer 3:

The manuscript entitled „Risk of Insulin Resistance and Metabolic Syndrome in Women with Hyperandrogenemia – a comparison between PCOS phe-notypes and beyond” by Borzan et al. is generally well written and structured in a reasonable way.

The main conception of the work seems reasonable. The effects of hyperandrogenism on metabolic abnormalities in patients with PCOS are widely known (as summarized e.g. by Sanchez-Garrido and Tena-Sempere (2020). Metabolic dysfunction in polycystic ovary syndrome: Pathogenic role of androgen excess and potential therapeutic strategies. Molecular metabolism, 35, 100937.). Also the question of the metabolic effects of hyperandrogenism in postmenopausal women is well-desribed in literature. In turn, the risk of development of the insulin resistance in hyperandrogenic premenopausal women needs a clear evaluation, which is the main aim of the submitted work.

Of course, some of the obtained results could be anticipated a priori. Nevertheless, it is important to evidently show the clinical results supporting the predicted relations, which is provided in this work.

We thank you for your kind words.

  1. Sometimes formatting looks strange especially by hyphenation e.g. charac-teris-tics, dis-eases.

We completely agree with you and thank you for pointing this out. Apparently this has occurred during the submission process, as the original manuscript of Dec. 25th did not show any of these changes. We have adapted the manuscript accordingly. We have corrected all hyphenated words we could find.

  1. There is a huge disparity between the sizes of control or test groups and the PCOS ones (for the data presented in Tab.1). One should make the inferences cautiously and ponder whether they are really statistically significant; especially when the data describing the phenotype A of PCOS (n = 392) are compared with almost all other groups (n<78) - in that case the sizes of control/other groups (except from PCOS B) are below 20% of the PCOS A group. I understand that the statistical tests confirmed the validity of the obtained relations between the investigated parameters. Still, the disproportions in sample sizes are can be a problematic issue.

Thank you for pointing out this topic. We agree with you that this is a serious issue that has to be addressed. We had already included this confounding factor into our limitations section in the 2

discussion and have further elaborated on the careful interpretation of our data as a result of this issue (page 12, paragraph 4, line 366).

  1. I think that the possible explanation of the observed relation between the free testosterone level and risk of developing metabolic type 2 diabetes and obesity should be provided taking into account the mechanistic link between them. In Discussion the Authors may provide the anticipated mechanism of how the androgens can imply the liability of patients to metabolic disorders. For example, the Authors can mention that androgens can downregulate hormone-sensitive lipase (HSL) and β2 adrenergic receptor expression etc.

We thank you for this suggestion and agree wholeheartedly that this topic should be expanded upon. As a result, we have dedicated three new paragraphs in the discussion to how androgen excess might lead to insulin resistance and other metabolic dysfunctions (page 11, paragraphs 5-7, lines 315-338). We hope that we have adequately addressed this concern.

  1. Could you comment on the relative amount of outliers in each group. Is it highest in PCOS A group? How can it be explained?

Thank you for these questions. We have addressed this topic: group A had the most frequent outliers in the boxplots (as can be seen in figure 2). This was already expected considering that phenotype A also had the largest number of participants (392), followed by 170 in phenotype B. In many cases, laboratory parameters have a skewed distribution, resulting in more extreme values on that side (usually the upper “normal” limit, which is often defined as the 95th percentile, meaning that 5 percent of all “normal” values are still outside that defined “normal” range). The more participants there are in a group, the more extreme values would be expected in that group, which is evident in our figure 2 boxplots. This is even more the case, when there are pathologically elevated parameter values in the data collection, which was the case in our PCOS cohort.

For the boxplots, outliers are defined as “all values above and below 1.5 times the interquartile range (IQR)”. Since the IQR only includes values between the 25th and 75th percentile, it is reasonable to assume that some values above the 95th percentile will not be included in the whiskers of the boxplot and will be defined as outliers or even extreme values.

To summarize, we believe that the number of outliers in our data (especially evident in our boxplots) are a result of how boxplot ranges are defined and how laboratory results are distributed. We hope that this comment answers your question satisfactorily.

We hope that our answers have addressed all comments satisfactorily and nothing more stands in the way of an acceptance of this manuscript. 3

Yours sincerely,

Obermayer-Pietsch and V. Borzan

Round 2

Reviewer 1 Report

The article can be accepted in its present form

Reviewer 2 Report

Thank you for the effort you put into the revision process.